

# A facial expression recognition network based on attention double branch enhanced fusion

Wenming Wang[1] and Min Jia[2]

[1] West Anhui University, Lu'an, Anhui, China
[2] Lu'an Hospital Affiliated to Anhui Medical University, Lu'an, Anhui, China

## ABSTRACT

The facial expression reflects a person's emotion, cognition, and even physiological or mental state to a large extent. It has important application value in medical treatment, business, criminal investigation, education, and human-computer interaction. Automatic facial expression recognition technology has become an important research topic in computer vision. To solve the problems of insufficient feature extraction, loss of local key information, and low accuracy in facial expression recognition, this article proposes a facial expression recognition network based on attention double branch enhanced fusion. Two parallel branches are used to capture global enhancement features and local attention semantics respectively, and the fusion and complementarity of global and local information is realized through decision-level fusion. The experimental results show that the features extracted by the network are made more complete by fusing and enhancing the global and local features. The proposed method achieves 89.41% and 88.84% expression recognition accuracy on the natural scene face expression datasets RAF-DB and FERPlus, respectively, which is an excellent performance compared with many current methods and demonstrates the effectiveness and superiority of the proposed network model.

## INTRODUCTION

Facial expressions are a direct carrier of human emotional changes and an indispensable nonverbal signal in social communication. Facial expression recognition (FER) is one of the main challenges in the field of emotional computing, with widespread applications in entertainment, marketing, retail, psychology, and other fields. Facial expression automatic recognition technology is widely used in human-computer interaction fields such as intelligent interrogation, autonomous driving, assisted medical care, consumer analysis, and clinical psychology. For example, *Sheng, Zhu-ying & Wan-xin (2010)* proposed an e-learning FER to capture students' real-time emotional states so as to adjust teaching strategies in a timely manner. In addition, FER has been applied in the field of rehabilitation to help and monitor patients and provide medical care by analyzing their emotions. *Comas, Aspandi & Binefa (2020)* combined peripheral physiological signals and

Corresponding author
Wenming Wang,
hellowwming@foxmail.com

facial expressions and proposed a multi-modal emotion recognition model based on deep learning technology, and applied the proposed method to the evaluation of anxiety treatment. *Al-Modwahi et al. (2012)* proposed improving surveillance systems by integrating FER to build a system that can detect malicious people from their facial expressions and report them to security personnel before they commit malicious acts. With the rapid development of machine vision research, how to obtain human emotional information through facial images has become a highly concerned research direction (*Cha, Choi & Im, 2020*; *Li et al., 2020*; *Wang et al., 2020b*).

There are many traditional methods for extracting facial expression features, among which the classic ones include the local binary pattern (LBP) method for extracting expression texture features (*Li & Li, 2019*), the histograms of oriented gradient (HOG) method for extracting expression edge features (*Bashyal & Venayagamoorthy, 2008*), and the scale invariant feature transform (SIFT) algorithm with a resistance to angle rotation and scale change (*Berretti et al., 2010*). Due to the small size of the facial images obtained through interception, as well as the large intra-class and inter-class differences among various expressions, manually designed feature extraction algorithms are easily affected by factors such as background, noise, and angle in practical applications. The recognition performance of the model is limited and the generalization ability is relatively insufficient.

Traditional FER widely uses machine learning methods to extract manually designed features, mainly relying on the knowledge and experience of designers, lacking sufficient reliability, and accuracy is also limited. Convolutional neural networks (CNN) can achieve end-to-end automatic feature extraction and classification, and the accuracy of FER can be significantly improved. Especially on laboratory datasets such as CK+ (*Lucey et al., 2010*), the accuracy of FER can reach over 95% (*Pang et al., 2018*; *Adil, Nadjib & Yacine, 2019*; *Yang, Ciftci & Yin, 2018*). However, as FER gradually moves towards practical scene applications, FER under natural conditions has attracted more attention. *Li et al. (2021)* proposed an LBAN-IL network for FER under natural conditions, which extracts sparse expression features while pruning redundant parameters, reducing model complexity; In addition, by increasing the amplitude of the vector to enhance the discriminative ability of the inter-class difference enhancement network, the accuracy of FER under natural conditions has been improved.

To avoid losing critical information, *Zhao, Liu & Wang (2021)* and *Weng et al. (2021)* adopt a multi-branch network to simultaneously learn global and local features and extract complementary information. *Chen et al. (2021)* and *Han et al. (2022)* enhance the network's ability to extract facial expression features by combining facial feature point detection. In addition, some methods also introduce attention mechanisms to suppress background interference (*Li et al., 2020*; *Wang et al., 2020a*; *Li et al., 2018*; *Wang et al., 2020b*), integrate deep learning features with traditional methods to enrich feature information (*Li et al., 2020*; *Zadeh, Imani & Majidi, 2019*; *Xi et al., 2020*), and significantly improve the accuracy of expression recognition. However, FER under natural conditions still faces many challenges, including complex background interference, large angle changes in posture, and uneven spatial information distribution. Its recognition accuracy

needs to be further improved (*Reddy, Savarni & Mukherjee, 2020*; *Sun et al., 2020*; *Woo et al., 2018*; *Lan et al., 2020*).

In response to the above issues, this article proposes a FER network based on attention double branch enhanced fusion. It focuses on the saliency of global and local features from the channel and spatial dimensions, respectively, as well as the use of a fusion strategy to enhance the complementarity between global and local features. The main contents and innovations are summarized as follows:

(1) A FER network based on attention double branch enhanced fusion is proposed. This network uses two parallel branches to capture globally enhanced features and local attention semantics and implements FER and classification through decision-level fusion strategies.

(2) To obtain enhanced global contextual information, the Convolutional Block Attention Module (CBAM) (*Woo et al., 2018*) attention mechanism is embedded in the global feature enhancement branch, which enhances global features from both channel flow and spatial flow perspectives, thereby improving the saliency and robustness of global features.

(3) To extract local detailed features, the Squeeze-and-Excitation (SE) (*Hu, Shen & Sun, 2018*) attention mechanism is embedded in the local feature attention branch to obtain local channel high-level semantics. Then, local features are fused and concatenated, and spatial high-level semantics are obtained through the spatial attention mechanism.

(4) The proposed method is experimented with and tested on the facial expression datasets of FERPlus (*Barsoum et al., 2016*) and RAF-DB (*Shan & Deng, 2018*) in natural scenes. The results show that the proposed method outperforms many advanced methods and has good accuracy.

The rest of this article is organized as follows. "Related Work" reviews the relevant literature on FER, with a special focus on attention mechanisms and feature fusion methods. "Methodology" describes the proposed method in detail, mainly including the design of the proposed network structure and loss function. Then, "Experiment and Result Analysis" introduces the experimental environment and experimental results and analyzes them in detail. Finally, "Conclusion" is the conclusion.

# RELATED WORK

Unlike traditional machine learning algorithms that heavily rely on complex rules, deep learning models can automatically learn typical features of various expressions (*Pham, Vu & Tran, 2021*). The CNN architectures commonly used for FER include AlexNet (*Krizhevsky, Sutskever & Hinton, 2017*), VGGNet (*Simonyan & Zisserman, 2014*), GoogleNet (*Szegedy et al., 2015*), and ResNet (*He et al., 2016*), which have been improved and optimized by researchers. *Zhao, Liu & Zhou (2021)* designed a Label Generation Generator to generate label distributions for training, using local feature extractors and channel space modulators to learn facial features. They developed a lightweight and robust facial expression recognition network (EfficientFace) for natural scenes. *Wang & Abisado (2023)* addressed the issues of insufficient generalization ability and low recognition

accuracy in children's expression recognition models. Using VGG16 as the baseline network, they introduced multi-scale convolutional modules and mixed attention mechanisms into the network and proposed a multi-scale Mixed Attention Mechanism Network (MMANet) for children's expression recognition. In terms of model design and optimization, the reference proposed an optimization-based deep learning model for early high-performance detection of solder paste defects on PCBs (*Sezer & Altan, 2021*). Reference proposed a novel robust hybrid classification model based on feature selection supported by swarm optimization for real-time high-precision classification of diseases of apple, grape, and tomato plants (*Yağ & Altan, 2022*).

Attention mechanism is a type of optimization module that has achieved significant results in the field of deep neural network research in recent years. The attention mechanism labels important features in an image by recalculating their weights to suppress background interference and identify regions with rich information required in the image. *Li et al. (2020)* combined LBP features with attention models for FER. *Wang et al. (2020b)* proposed a regional attention network (RAN) that incorporates self-attention modules and relational attention modules to learn the attention weights of each region, improving the robustness of FER under occlusion and pose changes. *Wang et al. (2020a)* proposed a self-cure network (SCN) that reduces the weight of uncertain images and the impact of uncertain images and incorrect labels on network training.

The fusion of multi-level features and the extraction of richer and more diverse facial expression features have also received a lot of attention. *Zadeh, Imani & Majidi (2019)* used Gabor filters to extract image edges and texture information and then used deep CNN to extract deep features, improving the training speed and accuracy of the model. *Xi et al. (2020)* used a grayscale co-occurrence matrix to extract eight different facial expression texture features at the front end of the model, fused them with the original image, and input them into a parallel network composed of CNN, residual networks, and capsule networks to extract precise features highly correlated with facial expression changes.

*Ni et al. (2022)* proposed a cross-modal attention fusion network to enhance the spatial correlation between global grayscale, LBP, and depth image features. This method adopts a global feature recognition method based on overall facial information. Due to ignoring the collaborative information of facial details such as eyebrows, eyes, and mouth, its recognition effect needs to be further improved. *Wadhawan & Gandhi (2022)* used 68 facial key points to detect 24 facial regions of interest and constructed five sub-networks and an overall network to extract features for each region and the overall feature. *Yu et al. (2020)* designed a global module to focus on overall facial features and utilized local modules to obtain fine-grained features of the region. *Zhao, Liu & Wang (2021)* used symmetric multi-scale modules to extract global features and introduced attention mechanisms to obtain local semantics. *Huang et al. (2021)* utilized the grid attention mechanism and self-attention mechanism to extract local and global features. To obtain global and local salient features with different fine-grained features, *Liu et al. (2022)* designed an adaptive multi-layer perceptual attention network by analyzing facial perception mechanisms and facial attributes.

Although these methods can capture effective global and local features for facial expression discrimination, they still have the following problems in natural scene applications: (1) Global features obtained based on a single channel flow or spatial flow not only have too single semantics, but also ignore global contextual information, and their saliency and robustness need to be further enhanced. (2) The local features extracted based on CNN make it difficult to highlight the salient regions of the features, and using a single channel or spatial attention mechanism is difficult to focus on the multi-dimensional semantics of the regions. The representation ability of local details needs further improvement. How to obtain globally enhanced features and local attention features from both channel and spatial perspectives will be the problem that this article needs to solve.

## METHODOLOGY

### Network structure

This article proposes a facial expression recognition network based on attention double branch enhanced fusion. This network is based on ResNet18 as the basic architecture, using two parallel branches to capture globally enhanced features and local attention semantics, and fusing the two branches through decision-level fusion to achieve expression classification. The network mainly consists of four parts: a pre-feature extraction module, a global feature enhancement branch, a local feature attention branch, and a decision-level fusion module. The overall structure of the network is shown in Fig. 1.

The pre-feature extraction module is used to extract intermediate facial features, which consists of a two-dimensional convolutional layer and three basic convolutional blocks. The structure of basic convolutional blocks is the basic block structure used in ResNet-18. A dual branch network performs subsequent processing on the pre-extracted facial feature map, thereby obtaining global and local features separately.

### Global feature enhancement branch

The global feature enhancement branch takes ResNet18 as the baseline network, takes the pre-extracted entire feature map as input, and embeds the CBAM attention mechanism before and after the last convolutional block in the ResNet18 network so that the network pays attention to both channel and spatial feature information from a global perspective. This method uses the CBAM attention mechanism to enhance the representation of face discriminative features in the global feature extraction part. The neural network first learns which features are key features through the channel attention module, and then uses the spatial attention module to learn where the key features are, strengthen the acquisition of image discriminative features, and perform adaptive refinement on the features. CBAM mainly consists of a serial structure of channel attention and spatial attention, as shown in Fig. 2. The importance of each channel in a network feature map is different. Channel attention can explore the dependency relationships between channel maps, extract the importance of each channel feature to key information by assigning weights, and selectively focus on information with high weight values.

The channel attention module compresses the input feature map F in the plane dimension through global average pooling and maximum pooling. Subsequently, the two

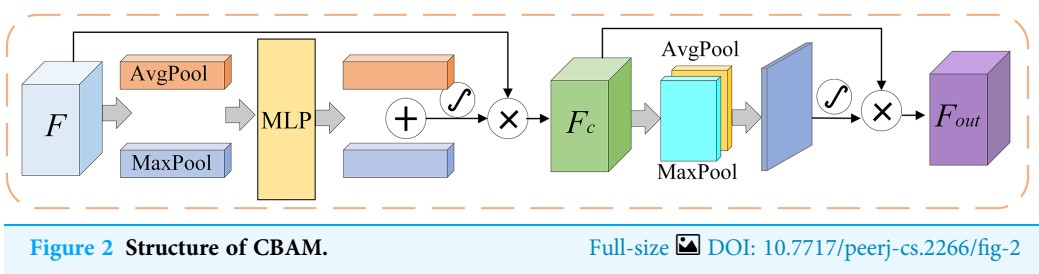

Figure 1 The structure of the proposed network model.

Figure 2 Structure of CBAM.

parts of the compressed features are respectively placed in a multi-layer perceptron (MLP) with a hidden layer for dimensionality reduction and dimensionality increase operations to extract the weight vector used to express the importance of the channel. The output features processed by the MLP are summed up and then activated by the sigmoid function to generate the final channel attention weight coefficient, as shown in the Formula (1). CBAM multiplies the weight coefficient $W_c$ obtained by the channel attention module with the input feature F to obtain a feature map $F_c$ containing more key information in the channel dimension.

The spatial attention module compresses the input feature map $F_c$ in the channel dimension through global average pooling and maximum pooling. Then, the two feature

maps are concatenated along the channel dimension and then passed through a convolution kernel activation function to generate the spatial attention weighted coefficient, as shown in Eqs. (3) and (4).

$$W_c = sigmoid(MLP(AvgPool(F)) + MLP(MaxPool(F))) \tag{1}$$

$$F_c = F \otimes W_c \tag{2}$$

$$F_m = Concat(AvgPool(F_c) + MaxPool(F_c)) \tag{3}$$

$$W_s = Sigmoid(Conv(F_m)) \tag{4}$$

$$F_{1out} = F_c \otimes W_s \tag{5}$$

In Eqs. (1)–(5) above, $F$ is the input feature map, $W_c$ is the channel weight, and $F_c$ is the feature map obtained after channel attention. Perform pooling operation on each feature point of $F_c$ to obtain two matrices. Then, concatenate the two matrices and obtain the weights $W_s$ for each spatial position through a convolutional layer and a sigmoid function. Finally, apply the weights to feature map $F_c$ to generate feature $F_{out}$ with enhanced spatial importance.

## Local feature attention branch

The global feature enhancement branch obtains globally enhanced features that reflect the overall information of the feature map, but it ignores the local details of the feature map. The expression details of local facial regions play an important role in FER, such as the mouth, eyebrows, eyes, *etc.*, which are more important than the nose, chin, and forehead (*Li et al., 2018*). Related studies have shown that attention mechanisms can effectively focus on regions of interest and highlight the role of important regions (*Chaudhari et al., 2021*; *Zhang & Li, 2022*).

The local feature attention branch directly divides the intermediate layer feature map into several non-overlapping local feature maps and guides the network to autonomously focus on local salient features through attention mechanisms, thereby learning more robust facial features. Firstly, divide the feature map $F \in \mathbb{R}^{C \times H \times W}$ output by the ResNet18 pre-feature extraction module into $P$ nonoverlapping region blocks, and input the region block feature $F_p$ into convolution to achieve dimensionality enhancement:

$$\{F_p\}_{p=1}^{P} = Split^s(F) \tag{6}$$

$$X_p = Conv_{3\times3}(F_p). \tag{7}$$

In Eqs. (6)–(7), $F_p \in \mathbb{R}^{256 \times 7 \times 7}$ represents the features of the p-th region after spatial partitioning, $X_p \in \mathbb{R}^{512 \times 3 \times 3}$ represents the features of the p-th region after convolutional dimensionality enhancement, and $Split^s(\cdot)$ represents the partitioning operation in the spatial dimension.

To highlight the significant information of channel semantics, $X_p$ is weighted through SE channel attention to obtain channel attention representation. SE attention is shown in Fig. 3, and the calculation is shown in Eqs. (8)–(10).

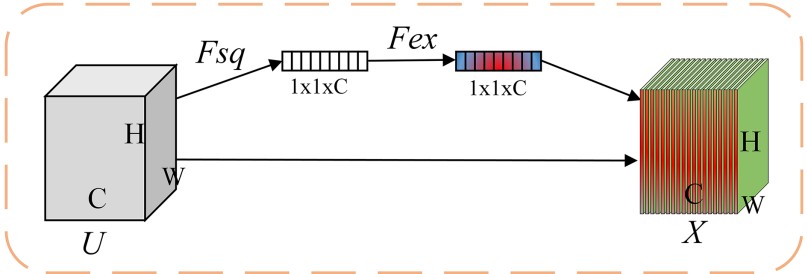

**Figure 3  Structure of SE attention.**

$$Z_c = F_{sq}(u_c) = \text{avgpool}_c(U) \tag{8}$$

$$S_c = F_{ex}(Z_c, W) = \sigma(g(Z_c, W)) = \sigma(W_2\delta(W_1 Z_c)) \tag{9}$$

$$\widetilde{X}_c = F_{scale}(U_c, S_c) = S_c \times U_c. \tag{10}$$

Although the channel attention mechanism characterizes local features from the channel dimension, it lacks attention to spatial information of local regions. Therefore, the p region blocks $X_p^c$ are concatenated into a feature block $X^c$, and then a spatial attention mechanism is embedded to enhance the spatial representation ability of local features.

$$X_p^c = SE(X_p) \tag{11}$$

$$X^c = Concat\left(\left\{X_p^c\right\}_{p=1}^p\right) \tag{12}$$

$$X_s^c = Spatial(X^c). \tag{13}$$

In the above equation, $X_p^c$ represents the feature map obtained by applying SE attention to the p-th region block. $Concat(\cdot)$ represents the feature concatenation operation in the spatial dimension. $X_s^c$ represents the feature map obtained through spatial attention. From the above process, it can be seen that this article embeds channel and spatial attention separately in the local feature attention branch, achieving the acquisition of local attention features from both channel and spatial dimensions, and improving the network's representation ability of local features.

Spatial attention, as shown in Fig. 4, is essentially searching for spatial information in the original image, transforming it into another space, and retaining key information. It weights the output for each position, enhances specific target areas, and enhances the feature expression ability of the target area.

$$F'_{avg} = AvgPool(F) \tag{14}$$

$$F'_{max} = MaxPool(F) \tag{15}$$

$$W_s = Sigmoid\left(Conv\left(Concat\left[F'_{avg}; F'_{max}\right]\right)\right) \tag{16}$$

$$F_s = W_s \otimes F. \tag{17}$$

Using the average pooling kernel to maximize the pooling of feature channel information in space, two feature maps $F'_{avg}$ and $F'_{max}$ were obtained. Further, use

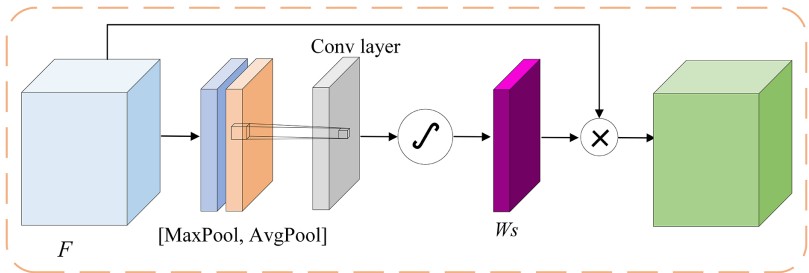

**Figure 4  Structure of spatial attention.**     

convolution and activation functions to extract features and obtain the weight matrix $W_s$. Finally, the feature map $F$ is multiplied by the weight $W_s$ to obtain the output feature $F_s$ for spatial attention.

### Loss function design

Capture global contextual information through global feature enhancement branches, and obtain local attention features through local feature attention branches. Perform global average pooling on both branches to obtain two 512-dimensional feature vectors. Then, they are passed through the fully connected layer to obtain two branches of expression category results, and finally, the final facial expression category is obtained through a decision-level fusion strategy. The loss function used in this article is cross-entropy loss, which can be expressed as:

$$L = -\frac{1}{N}\sum_{i=0}^{N-1} log \frac{e^{W_{y_i}^{(k)T} v_i^{(k)} + b_{y_i}^{(k)}}}{\sum_{j=0}^{C-1} e^{W_{y_j}^{(k)T} v_j^{(k)} + b_{y_j}^{(k)}}}. \tag{18}$$

Among them, $N$ represents the number of samples, and $C$ represents the number of expression categories. Both branches use cross-entropy loss as the loss function, and the losses generated by the two single branches are fused. The final loss function in this article can be expressed as:

$$L_{loss} = \lambda L_{global} + (1-\lambda)L_{local}. \tag{19}$$

Among them, $L_{global}$ represents the loss function of the global feature enhancement branch, and $L_{local}$ represents the loss function of the local feature attention branch. $\lambda \in [0, 1]$ is a hyperparameter used to balance the importance of global and local features.

## EXPERIMENT AND RESULT ANALYSIS

### Datasets

To analyze the expression recognition performance and generalization ability of the network proposed in this article, experiments were conducted on the facial expression datasets RAF-DB and FERPlus, respectively. Each image in the RAF-DB dataset is annotated by 40 well-trained programmers. The labels of this dataset contain both seven basic expression categories and mixed expressions. For basic expression category labels,

there are 12,271 facial images used for training and 3,068 facial images used for testing. FERPlus is a real scene expression dataset that has been re-labeled on the FER2013 dataset, and these images have been re-labeled as 10 categories of extremely imbalanced expressions. In this article, we chose to add context category expressions to seven basic expressions for the experiment. It includes 28,236 images as the training set and 3,137 images as the test set. The details of the dataset are shown in Tables S1 and S2. All images in the dataset are scaled to 224 × 224 pixels, and data enhancement methods such as random cropping, horizontal flipping, and erasing are used to avoid overfitting.

## Experimental environment

The experimental operating system is Ubuntu 18.04, with Python 3.9, CUDA11.3, and Python 1.12.0 as the basic environments for writing and running programs. The training uses NVIDIA GeForce RTX 2080Ti as the GPU. All input images are reshaped to 224 × 224 pixels. The experiment uses ResNet18 as the baseline network of the model. To evaluate the model fairly, ResNet18 is pre-trained on the MSCeleb-1M (*Guo et al., 2016*) face recognition dataset. The fused cross-entropy loss function was used and the SGD optimizer was used to optimize the model. The initial learning rate was set to 0.01, the total number of epochs was set to 100, and the decay was performed every 10 epochs with a decay rate of 0.1. The specific parameter settings are shown in Table S3.

## Result and analysis

### Model training and validation curve analysis

This article analyzes the accuracy and loss curves of the training and validation sets on the RAF-DB and FERPlus datasets. As shown in Fig. 5, it can be seen that 100 epochs are iterated on each of the two datasets.

As the number of iterations increases, the overall training accuracy curve is on the rise, indicating that the training accuracy is constantly improving. The loss value on the training set is continuously decreasing, indicating that the model is continuously learning and optimizing. In a relatively short learning period, the model converges faster. After the 20th epoch, the model tends to reach a stable state and achieves the accuracy of optimal performance. In addition, as the number of batches increases, the accuracy and loss values of the validation set are continuously optimized, and there is no overfitting or underfitting phenomenon in the network. Therefore, it is verified that the network and loss function constructed in this article have good generalization ability.

### Confusion matrix analysis

To more intuitively analyze the classification results of various categories of facial expressions on the dataset, this article presents the confusion matrix of the network on two datasets, as shown in Fig. 6.

From Fig. 6, it can be seen that for the seven expression categories on the RAF-DB dataset, the recognition difficulty of fear and disgust expressions is the highest, with accuracy rates of 64% and 65%, while the recognition accuracy of other expressions is above 85%, indicating a high recognition accuracy. Fear expressions are most likely to be misjudged as surprised and sad expressions, mainly because these three expressions have

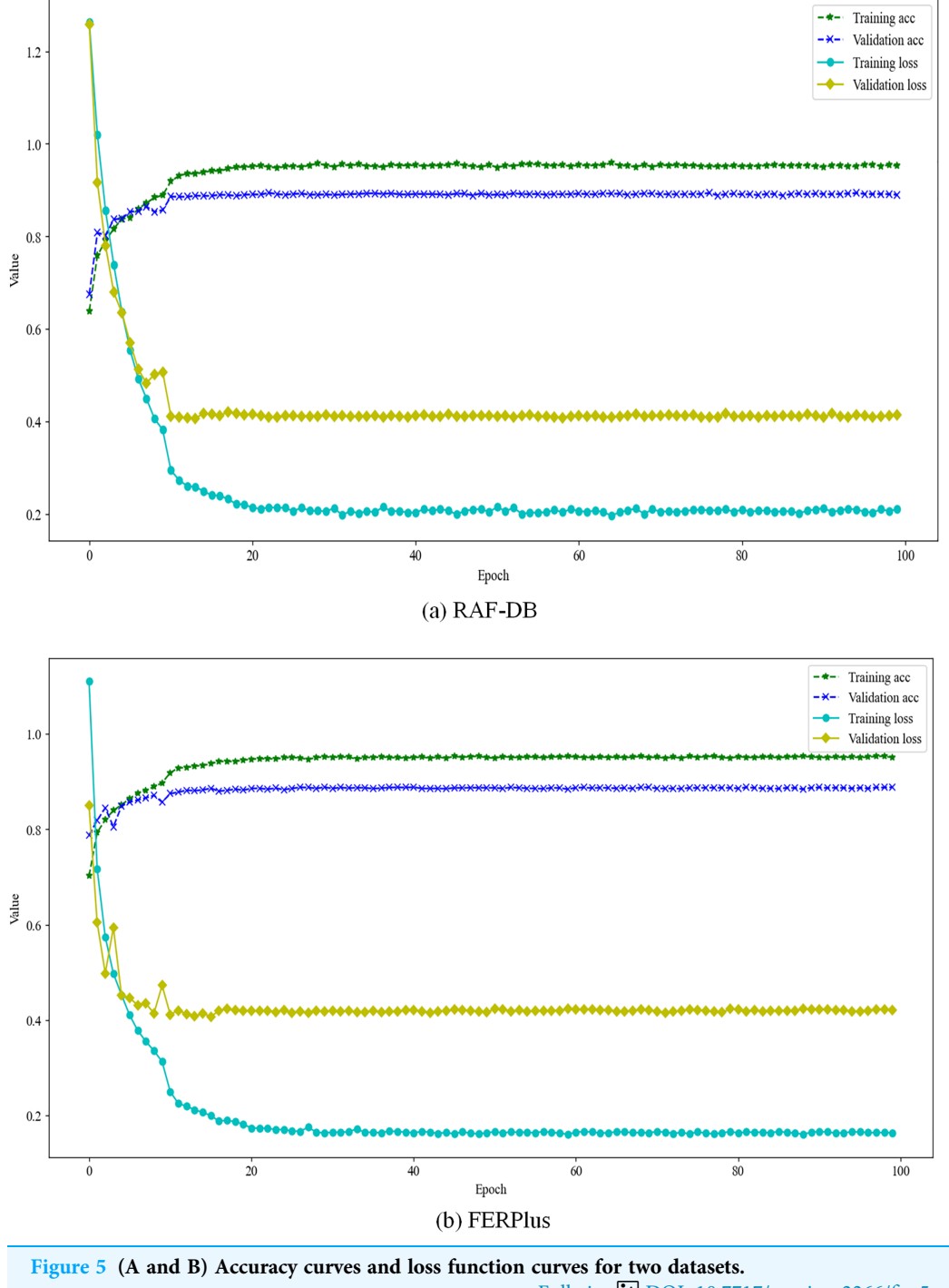

(a) RAF-DB

(b) FERPlus

**Figure 5 (A and B) Accuracy curves and loss function curves for two datasets.**

similar characteristics such as furrowed eyebrows and slightly open mouth, which can easily confuse. Happy expressions are easier to recognize compared to other expressions, with an accuracy rate of up to 95%. This is because happy expressions have significant features that are easier to recognize, such as curled corners of the mouth, slightly open mouth, and eyelid contraction.

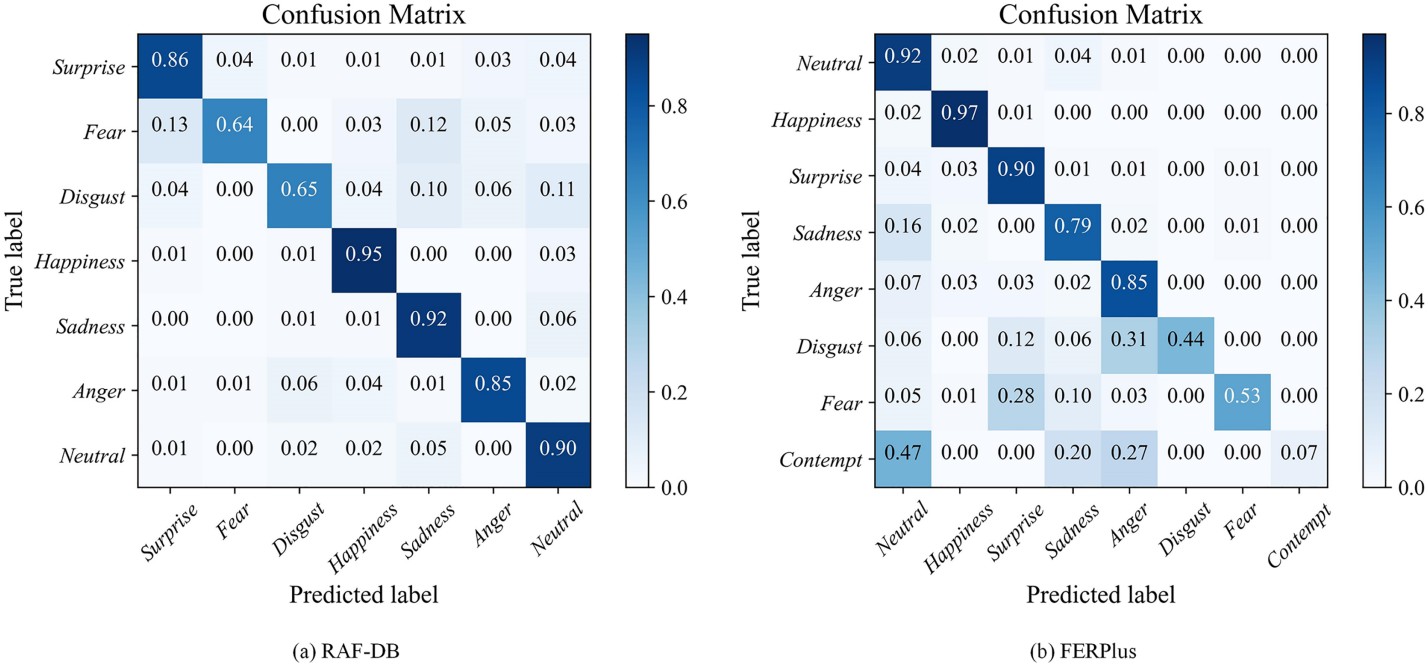

**Figure 6 The confusion matrix of the proposed model on two datasets.**

For the eight expression categories on the FERPlus dataset, the method proposed in this article has high accuracy in recognizing neutral, happy, and surprised expressions, all of which are over 90%. These expressions have obvious facial change characteristics, indicating that the method proposed in this article effectively extracts the characteristic changes of facial expressions and correctly recognizes them. However the recognition accuracy of the three expressions of disgust, fear, and contempt is relatively low. The probability of misjudging a disgust expression as an angry expression is 31%, the probability of misjudging a fear expression as a surprised expression is 28%, and the probability of misjudging a contempt expression as a neutral expression is 47%. On the one hand, it is due to the similar characteristics of these negative expressions, most of which have features such as downward corners of the mouth and furrowed brows. Facial key points usually only have slight changes, and the proportion of misclassification is high, which is easy to confuse and difficult to distinguish. On the other hand, by analyzing the data distribution of the FERPlus dataset (Fig. 7), it was found that in the FERPlus training and validation sets, the proportion of sample data for the three expressions of disgust, fear, and contempt was the least, and the sample size for the two expressions of disgust and contempt only accounted for 0.5% of the total sample size. The training samples were too small, and the distribution of sample data was severely imbalanced, resulting in insufficient network feature extraction, which also affected the recognition results of expressions.

### Evaluation through precision, recall, F1-score, and specificity

The performance of the proposed network model will be evaluated by precision, recall, F1-score and specificity. These metrics help to better determine the recognition effectiveness

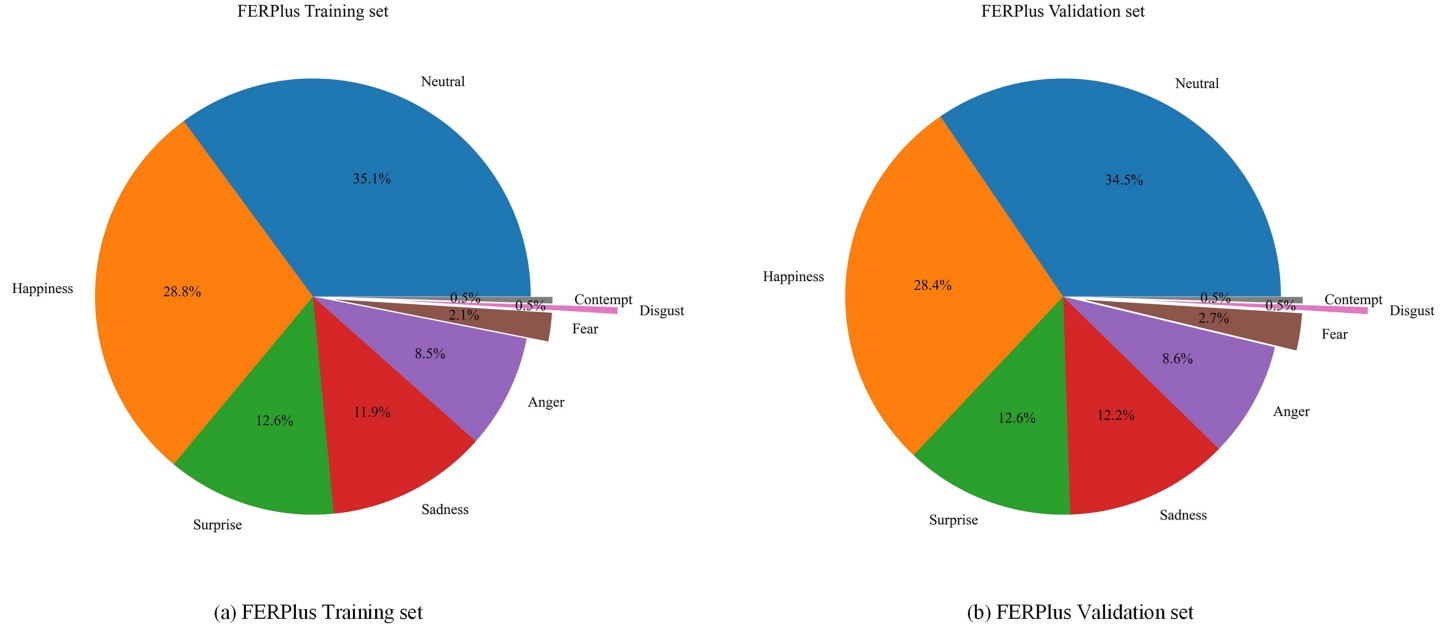

FERPlus Training set

FERPlus Validation set

(a) FERPlus Training set

(b) FERPlus Validation set

**Figure 7 (A and B) The data distribution of the FERPlus dataset.**

of the multi-classification problem, as shown in Tables S4 and S5. Precision refers to the proportion of images that belong to the category among all images identified as that category. It is used to measure the proportion of instances predicted by the model as positive examples, that is, the proportion of samples predicted by the model as true positive to the number of samples predicted as positive. Recall refers to the proportion of all images belonging to the category that are correctly identified, that is, the proportion of the number of images correctly classified as the category to the total number of images in the category. It is used to measure the proportion of samples correctly predicted by the model as positive. Specificity refers to the proportion of the number of images that do not belong to the category in the recognition result to the number of images that do not belong to the category. As can be seen from Tables S4 and S5, on the FERPlus dataset, the precision of expressions such as contempt is the lowest, while the precision of expressions of other categories is above 0.78. On the RAF-DB dataset, the precision of expressions such as disgust and fear is low, while the precision of expressions of other categories is above 0.82. The reason is that compared with other expressions, the above three expressions are all negative emotions, which are difficult to collect and the number of samples is small. In addition, the facial change characteristics of these three expressions are similar, which is easy to confuse and makes not easy to train and recognize network models.

### Receiver operating characteristic curve

To visually observe the recognition ability of the proposed network model for different expressions, the receiver operating characteristic (ROC) curve is plotted as shown in Fig. 8. The continuous line represents the ability of the network to recognize different expression
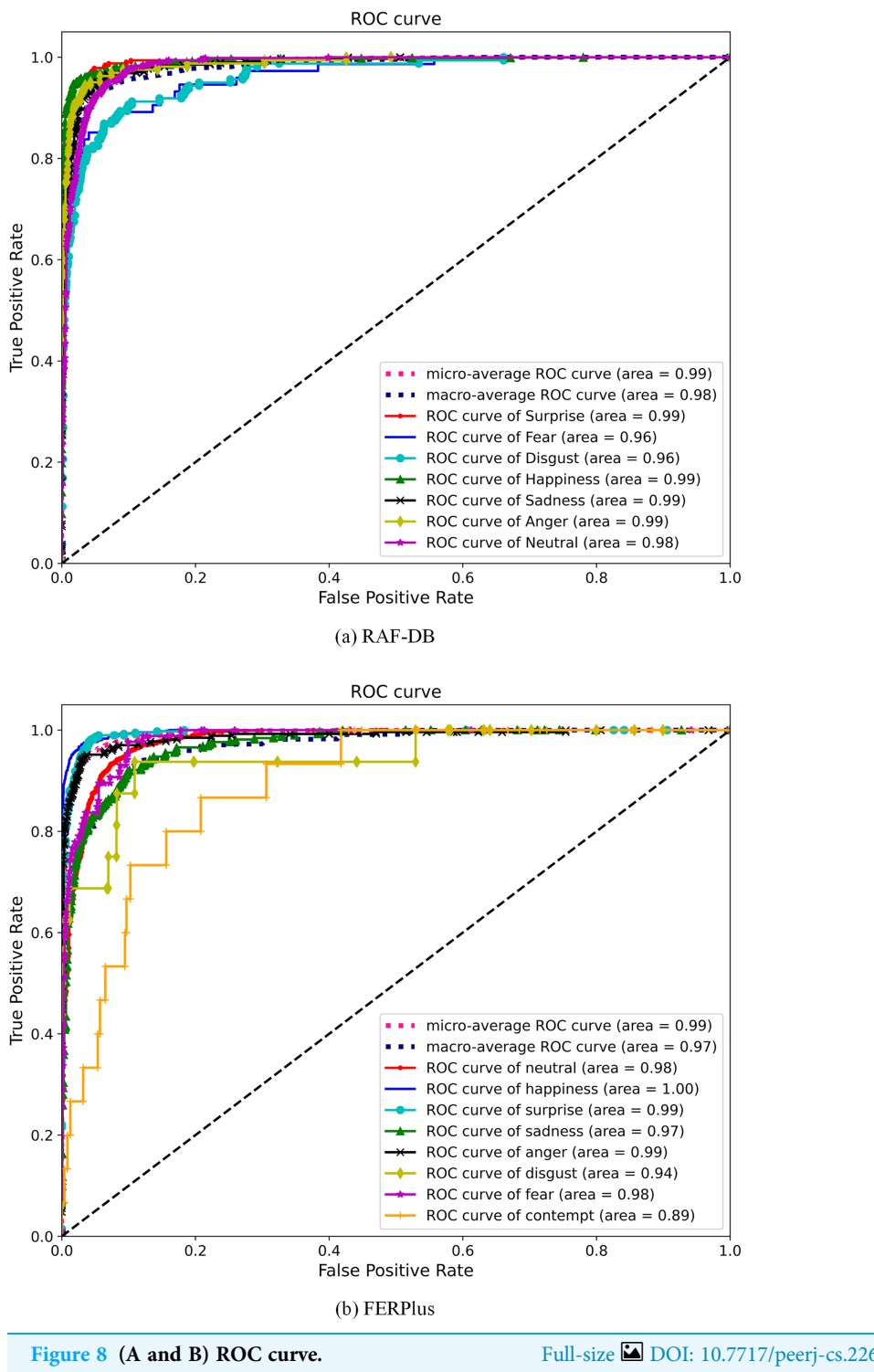

(a) RAF-DB

(b) FERPlus

**Figure 8** **(A and B) ROC curve.**

categories. The discontinuous lines are the Micro-Average ROC curve and the Macro-Average ROC curve. The higher the two are, the better the classification achieved by the network. The larger the area under curve (AUC) of the ROC curve, the better the performance of the model. From Fig. 8, we can see that the performance of the proposed
network model in the RAF-DB dataset is better than that in the FERPlus dataset. This is because the FERPlus dataset uses web crawler technology to automatically crawl facial expression images from web pages, which may contain partially occluded, missing, and blurred images of the face, resulting in an impure dataset. The RAF-DB dataset is labeled by professional staff, making the data quality higher. As shown in Fig. 8A, on the RAF-DB dataset, the AUC values of five expressions reached 0.99, and the AUC values of all expressions were above 0.96. On the FERPlus dataset, the AUC value of one expression (Happiness) reached 1, the AUC values of three expressions reached 0.99, and the AUC values of all expressions were above 0.89. From the ROC curve in Fig. 8, it can be seen that the proposed network model achieved good recognition results on both datasets, proving its good performance.

### Fusion strategy analysis

There are two commonly used fusion methods: feature-level fusion and decision-level fusion. Feature-level fusion is to concatenate the feature vectors obtained from the two branches into a joint feature vector, and then train a classifier to obtain the final classification result. Decision-level fusion is to add the recognition results of the two branches. To explore the impact of different fusion strategies, verification was carried out on two data sets, and the experimental results are shown in Table S6. For the network proposed in this article, the global feature enhancement branch and the local feature attention branch are face features extracted from two different ways, and their complementarity at the feature layer is weak. Therefore, it is more effective to adopt a decision-level fusion strategy.

To illustrate the role of the global feature enhancement branch and the local feature attention branch in the proposed network, this article discusses the impact of different weights $\lambda$ on the expression recognition performance on the datasets RAF-DB and FERPlus. As shown in Fig. 9, $\lambda$ is a hyperparameter to balance the two parts of the loss function. When $\lambda = 0$, the network only focuses on the local feature attention branch and lacks the compensation information of the global feature enhancement branch. The network performance is low on both datasets. As $\lambda$ gradually increases, the network integrates global features and local features, and the network performance shows an upward trend on both datasets. When $\lambda = 0.6$, the network performance reaches a peak, and the expression recognition rates on the datasets RAF-DB and FERPlus are 89.41% and 88.84%, respectively. As $\lambda$ continues to increase, the network gradually ignores the role of the local feature attention branch, and the network performance shows a downward trend. When $\lambda = 1$, due to the lack of compensation information of the local feature attention branch, the network performance is low. The above experimental results show that the proposed global feature enhancement branch and local feature attention branch can obtain semantic features at different levels and improve network performance through information complementation, and the global feature enhancement branch has a more significant impact on network performance.

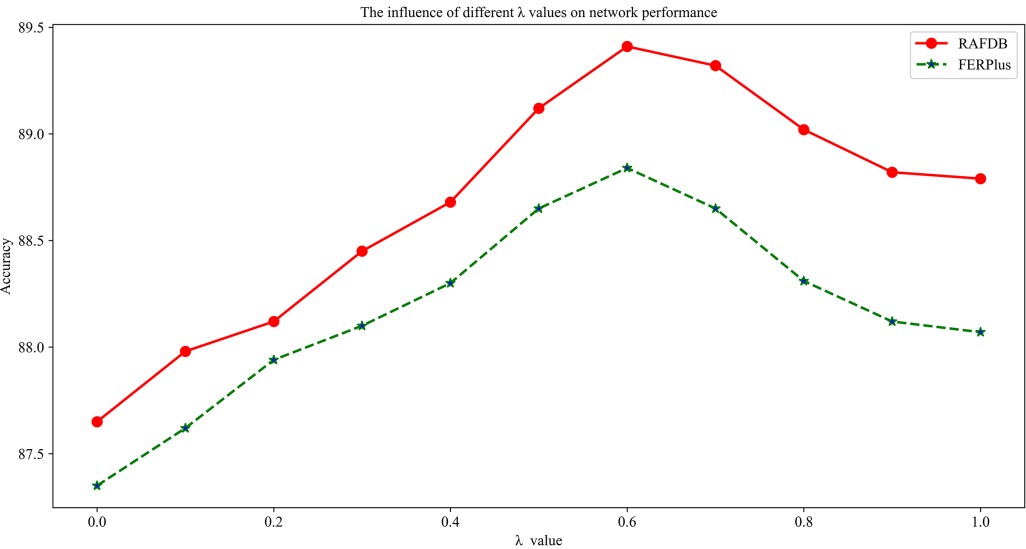

**Figure 9 Influence of different λ values on network performance.**

## Computational complexity analysis

Table S7 compares the proposed method with the state-of-the-art methods in terms of model size and inference time. The proposed method uses 16.33 M and 1.87 G parameters and FLOPs, respectively. Compared with other methods, it achieves better results on the dataset with similar parameters and computational complexity. This shows that the proposed method can achieve state-of-the-art FER performance with moderate resource consumption.

## Visualization analysis

To visually verify the roles of the global feature enhancement module and the local feature attention module, this article visualizes the convolution process using Grad-CAM (*Selvaraju et al., 2017*) technology. Grad-CAM obtains the importance of pixels in convolutional layers by calculating gradients and displays the features learned by the network through constructing heat maps. Pixel regions with higher model attention will have higher brightness.

As shown in Fig. 10, the first two rows show the visualization effect of the global feature extraction and global feature enhancement branches, while the third row shows the visualization effect of the local feature attention branch. The first and second columns in Fig. 10 are respectively expressions of surprise and fear, which are generally characterized by an open mouth and wide eyes. From the first row of the first and second columns, it can be seen that the global feature focuses on the features below the eyes in the entire facial area, and the global enhancement feature strengthens the global feature but ignores the changes in the local key area of the eyes. The local enhancement feature extracts the changes in the local area of the eyes but ignores the key features of the entire face. The fourth column is a happy expression. Both the global feature and the global enhancement feature focus on the changing area of the entire facial expression, and the local

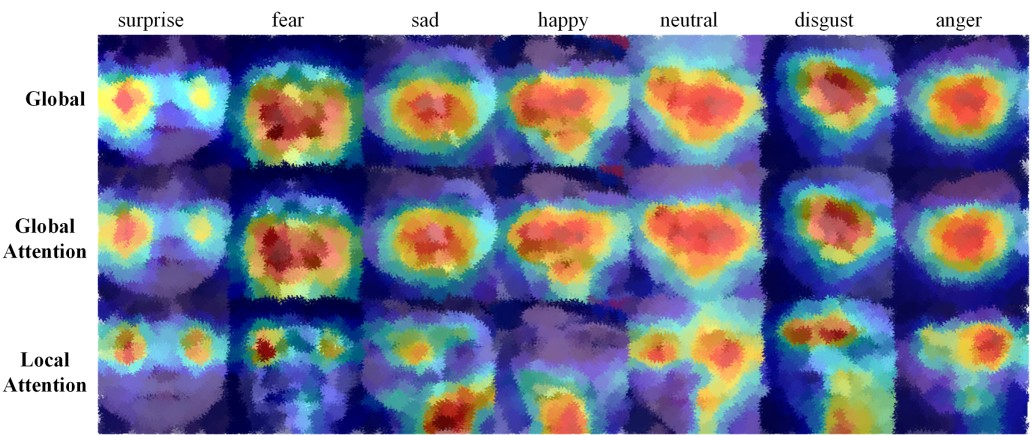

**Figure 10 Visualization of global feature extraction module, global feature enhancement branch, and local feature attention branch.**

enhancement feature extracts the local key features of the open mouth. By comparison, it can be seen that the global feature focuses on the features of the entire facial expression area. By embedding the attention mechanism into the global feature extraction to construct a feature enhancement branch, the feature areas that contribute more to expression classification, such as eyes and mouth, are enhanced, which improves the learning of discriminative feature areas and suppresses the acquisition of useless information.

Compared with the global feature enhancement branch that focuses on the entire facial region, the local feature attention branch can concentrate on the local area of the face. Facial expression features are mainly reflected in several local areas, such as eyebrows, mouth, nose wings, *etc*. Different types of expressions will have different expressions in these local areas. As shown in Fig. 10, the local feature attention branch can better focus on recognizable facial regions such as the eyes, mouth, and eyebrows, thus enabling the network to have better recognition performance. The local feature attention branch can pay more attention to the effective local feature regions that play an important role in classification, enabling the model to capture effective features for classification.

## Comparison with existing methods

To verify the effectiveness and superiority of the method proposed in this article, some typical CNN-based FER methods in recent years were selected for comparative experiments, and the results are shown in Tables S8 and S9.

From Table S8, it can be seen that our method outperforms many advanced methods, achieving an accuracy of 89.41% on the RAF-DB dataset. Compared with the ResNet18 baseline network, it has improved by 6.68%, and compared with mainstream advanced methods such as LANet (*Ma, Celik & Li, 2021*), MAPNet (*Ju & Zhao, 2022*), MFNet (*Gong et al., 2022*), SCAN+CC (*Gera & Balasubramanian, 2021*), it has increased by 2.71, 2.15, 0.88, and 0.39 percentage points, respectively. Overall, the proposed method has good competitiveness on the RAF-DB dataset.

From Table S9, it can be seen that the proposed method achieved an accuracy of 88.84% on the FERPlus dataset, which was improved by 0.74, 0.42, 0.12, and 0.03 percentage points compared to methods such as SpResNet-ViT (*Gao, Li & Zhao, 2022*), IPD-FER (*Jiang & Deng, 2022*), PACVT (*Liu, Hirota & Dai, 2023*), and VTFF (*Ma, Celik & Li, 2021*), respectively. Among the several advanced algorithms compared, the highest performance was achieved, indicating the excellent performance of the proposed method.

On two different representative datasets, the proposed network model can achieve the best recognition performance. The main reasons are:

1) The global feature enhancement branch is used to learn the feature information of the entire face and the local feature attention branch is used to extract the local key features of facial expressions.

2) The attention mechanism is used in both the global and local branches to emphasize the areas related to facial expressions and suppress the irrelevant background areas, thereby improving the expressiveness of important facial features.

3) The fusion and complementarity of global and local information are achieved through decision-level fusion, making the recognized features more complete, so the recognition accuracy is higher than other algorithms.

## CONCLUSION

This article proposes a FER network based on attention double branch enhanced fusion to solve the problems of insufficient feature extraction, loss of local key information, and low recognition accuracy in complex FER. On the one hand, the proposed network constructs a channel-space global feature enhancement structure to obtain channel semantics and pixel-level spatial semantics, realizes the mining of global features from two perspectives, channel flow, and spatial flow, and enhances the discriminability of global features. On the other hand, by locally dividing the feature map and embedding channel attention, and then splicing and embedding spatial attention, the saliency of local features is improved. In addition, the weighted fusion and mutual collaboration of global enhancement features and local attention semantics are achieved through decision-level fusion, which improves the robustness of expression recognition in natural scenes.

The accuracy curve and loss curve on the training and validation sets verify that the network and loss function constructed in this article have good generalization ability. The confusion matrix further analyzes that the model has good accuracy in recognizing various expressions. The recognition effect of the network on multi-classification problems is analyzed by precision, recall, and specificity, and the recognition ability of the proposed network model for different expressions is observed by the ROC curve. In addition, the fusion strategy and computational complexity adopted by the network model in this article are analyzed. In addition, the output feature maps of the global feature extraction module, the global feature enhancement branch, and the local feature attention branch are visualized. Experimental results show that the expression recognition accuracy of the proposed network on the RAF-DB and FERPlus datasets is 89.41% and 88.84%

respectively, which is superior to many current advanced methods of FER, demonstrating that the proposed model has good advancement and robustness.

Although the proposed method exceeds some mainstream methods, its accuracy still needs further improvement. In future work, it is of great significance to design more robust and lightweight network models to improve facial expression recognition in complex environments and enhance its application value in many fields.

### Funding
This work was supported by the Natural Science Key Project of West Anhui University (No. WXZR202310). The funders had no role in study design, data collection and analysis, decision to publish, or preparation of the manuscript.

### Grant Disclosures
The following grant information was disclosed by the authors:
Natural Science Key Project of West Anhui University: WXZR202310.

### Competing Interests
The authors declare that they have no competing interests.

### Author Contributions
- Wenming Wang conceived and designed the experiments, performed the experiments, performed the computation work, authored or reviewed drafts of the article, and approved the final draft.
- Min Jia conceived and designed the experiments, analyzed the data, performed the computation work, prepared figures and/or tables, authored or reviewed drafts of the article, and approved the final draft.

### Data Availability
The RAF-DB Dataset is available at: http://www.whdeng.cn/RAF/model1.html
The FERPlus Dataset is available at Kaggle: https://www.kaggle.com/datasets/lixiqi/ferplus
The code is available at GitHub: https://github.com/WangReview/FER

### Supplemental Information
Supplemental information for this article can be found online at http://dx.doi.org/10.7717/peerj-cs.2266#supplemental-information.

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
