# Peer review of "A facial expression recognition network based on attention double branch enhanced fusion"

_PeerJ Computer Science, doi:10.7717/peerj-cs.2266_

## Round 0.1 · original submission · Major Revisions

Based on the reviewers comments, the manuscript must be revised accurately.

**Language Note:** The review process has identified that the English language must be improved. PeerJ can provide language editing services - please contact us at [email protected] for pricing (be sure to provide your manuscript number and title). Alternatively, you should make your own arrangements to improve the language quality and provide details in your response letter. – PeerJ Staff

Reviewer 1 ·

Basic reporting

The manuscript entitled “A facial expression recognition network based on Attention Double Branch Enhanced Fusion” has been investigated in detail. The paper introduces a facial expression recognition network based on attention dual-branch enhanced fusion to address challenges such as background interference and uneven spatial information distribution in facial expression recognition. The proposed network is based on the ResNet18 architecture and uses two parallel branches to capture global enhanced features and local attention semantics. These branches are then fused through decision-level fusion for expression classification. Experimental results show that the proposed method achieves expression recognition accuracy of 89.41% on the RAF-DB dataset and 88.84% on the FERPlus dataset. There are some points that need further clarification and improvement:
1) The proposed method's contribution is not sufficiently distinguished from existing facial expression recognition networks. The approach seems to rely heavily on the ResNet18 architecture and dual-branch attention, which have been utilized in other studies. More discussion on the novelty and uniqueness of the approach is needed.
2) The paper does not provide a detailed explanation of the attention mechanism used in the dual-branch approach. The mechanism's impact on the network's performance and how it specifically addresses the issues of background interference and uneven spatial information distribution should be clarified.

Experimental design

3) While the paper mentions achieving high accuracy compared to other advanced methods, the comparative analysis lacks depth. There should be a more comprehensive comparison with state-of-the-art methods, including a discussion of the strengths and weaknesses of the proposed model in relation to others.
4) The decision-level fusion approach is not thoroughly justified. The paper should provide more information on why this fusion method was chosen over other potential fusion strategies and its benefits for facial expression recognition.
5) The paper does not adequately discuss the evaluation process, including potential sources of bias, validation methods, and the potential impact of training data distribution on performance. More details on the evaluation methodology are needed.

Validity of the findings

6) The paper's claim of high accuracy on the RAF-DB and FERPlus datasets seems overly confident without sufficient discussion of possible limitations or challenges faced during the study.
7) The paper does not provide enough information for other researchers to replicate the study. More details on the implementation, including code, data preprocessing steps, and hyperparameters, are needed to ensure reproducibility.
8) “Discussion” section should be added in a more highlighting, argumentative way. The author should analysis the reason why the tested results is achieved.
9) The authors should clearly emphasize the contribution of the study. Please note that the up-to-date of references will contribute to the up-to-date of your manuscript. The studies named- “Detection of solder paste defects with an optimization‐based deep learning model using image processing techniques; Artificial intelligence-based robust hybrid algorithm design and implementation for real-time detection of plant diseases in agricultural environments”- can be used to explain the methodology in the study or to indicate the contribution in the “Introduction” section.
10) It will be helpful to the readers if some discussions about insight of the main results are added as Remarks.

Additional comments

The paper proposes a facial expression recognition network based on attention dual branch enhanced fusion with the aim of improving accuracy under natural conditions. While the study presents promising results, there are several areas that need improvement, including a more detailed explanation of the attention mechanism, better comparative analysis, and more transparency in the methodology. Additionally, the paper should address potential limitations and ethical considerations related to facial expression recognition. Overall, the study requires significant revision and additional evidence to support its claims.

Reviewer 2 ·

Basic reporting

The manuscript entitled “A facial expression recognition network based on Attention Double Branch Enhanced Fusion” has been investigated in detail. After reading the paper, the reviewer has the following concerns:
1) The proposed network architecture relies heavily on ResNet18, a well-established model in computer vision, with limited innovation in the attention mechanism or fusion approach.
2) The paper lacks comprehensive comparisons with state-of-the-art methods beyond the reported accuracy rates. More detailed evaluations against diverse and recent benchmarks are necessary to establish the method's competitiveness.

Experimental design

3) The paper lacks detailed explanations of the fusion process at the decision level. Further elaboration on how the branches interact and the impact of attention mechanisms is necessary for reproducibility and understanding.
4) Although the proposed model is tested on RAF-DB and FERPlus datasets, the use of a broader range of datasets would provide more robust and generalizable results. The current dataset selection may not adequately represent diverse real-world scenarios.

Validity of the findings

5) The paper primarily focuses on accuracy; however, additional metrics such as precision, recall, and F1-score should be considered for a more comprehensive evaluation of the model's performance.
6) The paper lacks discussion on potential challenges in real-world applications of the model, such as computational cost and deployment considerations.

Additional comments

The manuscript entitled “A facial expression recognition network based on Attention Double Branch Enhanced Fusion” has been investigated in detail. After reading the paper, the reviewer has the following concerns:
1) The proposed network architecture relies heavily on ResNet18, a well-established model in computer vision, with limited innovation in the attention mechanism or fusion approach.
2) The paper lacks comprehensive comparisons with state-of-the-art methods beyond the reported accuracy rates. More detailed evaluations against diverse and recent benchmarks are necessary to establish the method's competitiveness.
3) The paper lacks detailed explanations of the fusion process at the decision level. Further elaboration on how the branches interact and the impact of attention mechanisms is necessary for reproducibility and understanding.
4) Although the proposed model is tested on RAF-DB and FERPlus datasets, the use of a broader range of datasets would provide more robust and generalizable results. The current dataset selection may not adequately represent diverse real-world scenarios.
5) The paper primarily focuses on accuracy; however, additional metrics such as precision, recall, and F1-score should be considered for a more comprehensive evaluation of the model's performance.
6) The paper lacks discussion on potential challenges in real-world applications of the model, such as computational cost and deployment considerations.
Overall, while the article presents a facial expression recognition network, the contributions are limited by the lack of novel approaches and insufficient methodological depth. The paper would benefit from a stronger focus on demonstrating advancements in performance and practical applications of the proposed method. In summary, due to the lack of key technical discussions, comparative studies, and computational complexity analysis, I cannot recommend this manuscript to be further considered in this journal.

Reviewer 3 ·

Basic reporting

This article introduces an approach: a facial expression recognition network based on attention dual branch enhanced fusion. This network leverages the ResNet18 architecture and employs two parallel branches to capture both global enhanced features and local attention semantics. Through decision-level fusion, these branches are then combined to achieve expression classification.
The following suggestions can improve the quality of the manuscript.

Please improve the technical language of the paper
and provide an outline of the paper at the end of the introduction section.
Novelty should be justified in a separate section. The proposed network structure appears to be heavily reliant on existing architectures, such as ResNet18, without introducing significant innovations or modifications to justify its novelty in the field of facial expression recognition

Experimental design

it lacks depth in explaining the significance of addressing this issue, such as real-world applications or implications for various fields like human-computer interaction, affective computing
Providing more clarity on the rationale behind the choice of ResNet18 architecture and the mechanisms of global enhanced feature extraction and local attention semantics would enhance the readers' understanding

the paper lacks detailed information on the training process, such as data augmentation techniques employed, hyper parameters tuning, and training/validation/test splits. Providing such details would enable reproducibility and facilitate comparisons with other methods
The description of the network structure and methodology lacks clarity and depth, making it difficult for readers to understand the rationale behind certain design choices and the overall functioning of the proposed system
there is limited discussion on other crucial metrics, such as precision, recall, and F1 score. A comprehensive evaluation of these metrics would offer a more thorough understanding of the model's performance.

Validity of the findings

Figure 1 shows an image, have you taken permission to use this image , otherwise please blur it.
Figures 12, 13 and 14 seems to be better changed as Tables , presently they have been pasted as figures.
The discussion on the confusion matrix analysis is relatively brief and lacks in-depth interpretation of the classification results. Further elaboration on the reasons behind misclassifications and strategies for improving model performance would enhance the depth of the analysis.
While the visualization of the global feature extraction and local feature attention branches is provided, the interpretation of these visualizations is superficial. A more detailed discussion on how these visualizations contribute to the model's decision-making process would provide valuable insights into the inner workings of the proposed network
The rationale behind selecting ResNet18 as the benchmark network and specific parameter settings is not adequately justified. Providing reasoning behind these choices would strengthen the methodological foundation of the study and enhance its credibility within the research community

Additional comments

The paper needs further improvement in experiment analysis and results elaboration. The technical language of the paper should also be updated by a fluent English speaker

·

Basic reporting

In the study, the authors conducted a study to increase the accuracy of facial expression recognition and made a suggestion.
The article is written in English and uses clear, concise, technically correct text. The article complies with professional standards of courtesy and expression.
Literature references, adequate field history/context provided.
Professional article structure, figures are appropriate. Data has been shared. The results of the study were tried to be expressed clearly with Tables and Figures.
--- the confusion of tables and figures. Table 1 is shown in Figure 12 in the study.

Experimental design

This is a study within the Scope of the Journal.
The research question could have been defined more clearly. The research question could be supported with references.In the related works section, it is necessary to define the research problem, and the problems solved based on the researched literature
The problem proposed to be solved in the study was carried out at an appropriate technical standard and the results were supported by figures.

Validity of the findings

The validity of the findings is proven. In the study, the data used for the Results are acceptable and ready for use.
The results are expressed appropriately. The data in the tables and figures have been interpreted correctly

Additional comments

The article should be accepted if the authors make some revisions, minor enough

---

## Round 0.2 · accepted · Accept

Based on the reviewer comments, the manuscript can be accepted.

Reviewer 1 ·

Basic reporting

All my comments have been thoroughly addressed. It is acceptable in the present form.

Experimental design

All my comments have been thoroughly addressed. It is acceptable in the present form.

Validity of the findings

All my comments have been thoroughly addressed. It is acceptable in the present form.

Reviewer 2 ·

Basic reporting

My comments were taken into account by the authors and the study was revised accordingly. The study is suitable for publication in its current form.

Experimental design

My comments were taken into account by the authors and the study was revised accordingly. The study is suitable for publication in its current form.

Validity of the findings

My comments were taken into account by the authors and the study was revised accordingly. The study is suitable for publication in its current form.

·

Basic reporting

Suggested corrections were made by the authors.

Experimental design

Suggested corrections were made by the authors.
Additions made in the responses to referees and editors are included in the new version.
"Global feature enhancement branch"
"Fusion Strategy Analysis"

Validity of the findings

Suggested corrections were made by the authors.
Additions made in the responses to referees and editors are included in the new version.
"Comparison with existing methods"

Additional comments

Suggested corrections were made by the authors.
Additions made in the responses to referees and editors are included in the new version.
The article meets the PeerJ criteria and should be accepted as is.